# Use of Ultrasound for Body Composition in Assessment in Pediatric Patients: Are There Still Challenges?

**DOI:** 10.3390/diagnostics15192472

**Published:** 2025-09-27

**Authors:** Patricia Miranda Farias, Amanda Matos Lima Melo, Aryanne Almeida da Costa, Valter Aragão do Nascimento, Arnildo Pott, Rita de Cássia Avellaneda Guimarães, Karine de Cássia Freitas

**Affiliations:** 1Graduate Program in Health and Development in the Central-West Region of Brazil, Medical School, Federal University of Mato Grosso do Sul, Campo Grande 79070-900, Brazil; patmiranda_nut@hotmail.com (P.M.F.); valter.aragao@ufms.br (V.A.d.N.); rita.guimaraes@ufms.br (R.d.C.A.G.); 2Integrated Multiprofessional Residency Program in Health Area of Concentration Intensive Care, Regional Hospital of Mato Grosso do Sul, Campo Grande 79084-180, Brazil; amandamlmelo@gmail.com (A.M.L.M.); aryannealme1@gmail.com (A.A.d.C.); 3Laboratory of Botany, Institute of Biosciences, Federal University of Mato Grosso do Sul, Campo Grande 79070-900, Brazil; arnildo.pott@gmail.com

**Keywords:** malnutrition, ultrasonography, body composition, pediatric intensive care unit

## Abstract

Patients who present nutritional risk upon hospital admission are more likely to have worse clinical outcomes. Evaluating muscle thickness with ultrasound is a predictor of muscle mass loss in pediatric patients. We reviewed the muscle mass loss detection through ultrasound to assess the body composition of pediatric patients. We found an association between muscle reduction, as detected by ultrasound, and the duration of mechanical ventilation, nutritional deficits in energy and protein intake, and the age-related skeletal muscle atrophy of the limbs. All studies reported a reduction in muscle thickness of more than 10% during hospitalization. There is a lack of standardization in muscle mass assessment protocols and established cut-off points in critically ill hospitalized children. Further studies are needed to establish an accurate and standardized analysis for monitoring muscle changes using ultrasound.

## 1. Introduction

Pediatric malnutrition can be defined as an imbalance between nutrient intake and physiological needs, leading to energy–protein and micronutrient deficiencies that negatively impact development and growth. It can be classified as acute (lasting less than 3 months) or chronic (lasting 3 months or more), with chronic malnutrition manifesting as growth deficits. Another concept is hospital-acquired malnutrition, which refers to nutritional imbalance acquired during hospitalization and can occur with or without preexisting malnutrition [1,2].

Worldwide, approximately 149.0 million children under the age of 5 suffer from stunted growth and 49.5 million suffer from wasting. Asia accounts for more than half of children with stunted growth (81.7 million, 54.8%). Only about a quarter of children with severe malnutrition received treatment for this disease [3].

The mechanisms capable of triggering or aggravating malnutrition may be related to starvation, malabsorption, the loss of nutrients, and the presence of inflammation that causes hypermetabolism. Malnutrition is a complication that can be characterized as secondary to a disease or condition or unrelated to the disease. In addition, for critically ill patients, it is known that the role of inflammation, presenting concomitantly with malnutrition, demonstrates a higher degree of deterioration in nutritional status compared with body composition at admission [1].

The inflammation described above causes metabolic, hormonal, and glycoregulatory changes. In the initial phase, there is skeletal muscle loss caused by the use of the gluconeogenesis pathway through the utilization of amino acids. Subsequently, there is a change in fat leading to lipolysis and ketogenesis. Concomitantly, insulin and growth hormone are affected, with changes that include a reduction in triiodothyronine, insulin, growth factor (IGF-1), and an increase in GH and cortisol [4]. Insulin and glucagon regulate the oxidation and synthesis of macronutrients in the short term. After that, in the long term, this mediation is carried out by growth hormone, thyroid hormones, catecholamines, and corticosteroids [5].

Levels of inflammatory cytokines such as TNF, IL-1, IL-6, and IL-12 alter the growth hormone–growth factor axis, resulting in atrophy and growth failure [5]. Immunity is also affected as a result of atrophy of the thymus, lymph nodes, and tonsils, leading to a decrease in T lymphocyte differentiation, affecting phagocytosis and reducing immunoglobulin A. For this reason, increases the susceptibility to invasive infections such as urinary, respiratory, gastrointestinal, and others [4].

Analysis of the nutritional status of hospitalized children reveals that approximately one in eight to nine children suffer from some degree of malnutrition. According to the World Health Organization (WHO) classification criteria, the prevalence of acute malnutrition reaches 9.6%, while chronic malnutrition affects approximately 13.6% of the pediatric hospital population. The lack of standardization for measurements and values for cut-off points and assessment of pediatric nutritional status is considered complex and interferes with the estimation of a reliable prevalence of malnutrition in that population [6].

The absence of screening, nutritional assessment, and prolonged fasting periods are factors that contribute to a negative impact on malnutrition in hospitalized patients. The repercussions for these patients include longer hospital stays, increased complications such as infections, immune dysfunction, delayed healing, reduced tolerance to treatment, muscle weakness, and delayed developmental [1,7].

Considering these repercussions, nutritional screenings aim to identify whether the patient is at nutritional risk. They use parameters such as diagnosis, changes in appetite and food intake, body weight loss, anthropometric measurements, and growth curves [8]. Patients who are at nutritional risk upon hospital admission are more likely to have worse clinical outcomes, such as longer hospital stays and higher hospital costs [9]. There is no good standard screening tool for assessing hospitalized pediatric patients. Among those used are Screening Tool for Risk on Nutritional Status and Growth (STRONG-kids), Screening Tool for the Assessment of Malnutrition in Pediatrics (STAMP), and Pediatric Yorkhill Malnutrition Score (PYMS) [10].

The STRONGKids screening tool is a subjective questionnaire that uses clinical criteria, dietary, and gastrointestinal aspects to identify nutritional risk, and is widely used in Pediatric Intensive Care Units (PICUs). This screening does not use anthropometric data, but its use favors a more complete assessment of the patient, correlating nutritional risk with the patient’s prognosis. Once nutritional risk is identified early, a more specific nutrition assessment is necessary [11].

Ultrasound has the advantage of being a practical, safe, simple method that is free from radiation risks. It allows for a more accurate measurement of muscle mass compared to other less accurate assessment methods. This is a measurement that is not significantly influenced by water balance, since the overall physical dimensions of muscle fibers remain unchanged under these conditions. For this reason, it is considered a reliable and useful parameter for identifying muscle changes and assessing body composition. Muscle mass is assessed by measuring the average thickness of the quadriceps femoris muscle. Disadvantages include the need for prior training to use the equipment and its average cost, although it is more affordable than other imaging methods [12].

Despite the recent number of studies published on the use of ultrasound for assessing children’s body composition, many uncertainties remain regarding the methodologies used. Further clarification is needed to understand the standards used for this type of population. This literature review is timely because muscle mass loss during hospitalization remains a significant challenge in the hospital setting. It is necessary to study these processes in depth and show the importance of detailed nutritional assessment, particularly because the child’s current weight is not always available. As we will see throughout the text, adequate standardization will allow the establishment of ideal cut-off points for the pediatric range. This study aimed to analyze the use of ultrasound in assessing body composition in pediatric patients, focusing on the identification and description of the methodological standardizations adopted in the scientific literature.

## 2. Materials and Methods

To illustrate the use of ultrasound and body composition assessment in pediatric patients, articles were selected that addressed methods of body composition assessment, history of ultrasound use, existing models, the assessment of lean mass, thickness, echogenicity, midpoint for measurement, and frequency.

The searches were conducted in the Medline (PubMed/MEDLINE), Virtual Health Library (VHL), and Science Direct databases, using the following keywords according to the Health Sciences Descriptors (DeCS) and Medical Subject and Headings Section (MESH): “ultrasonography,” “ultrasound,” “point-of-care ultrasound assessment,” “body composition”, “quadriceps muscle”, “diagnostic imaging”, “nutrition assessment”, “malnutrition”, “muscle loss”, “muscular atrophy”, “pediatric intensive care unit”, “pediatric critical patient”, with boolean operators “AND” or “OR” for combinations of the established descriptors.

As inclusion criteria, studies in English and Portuguese were selected, covering the age range from 0 to 18 years. Different methodological designs were considered eligible, including original studies, narrative reviews, integrative reviews, systematic reviews, and meta-analyses. Regarding the date of publication, journals published in the last 10 years (2015–2025) were included due to the scarcity of studies on the subject. The following were excluded: studies with editorial designs, reviews, letters, undergraduate theses, dissertations, and theses; non-peer-reviewed articles such as comments and conference abstracts; studies in other languages; studies that did not cover the established age range; experimental studies; studies with cadavers; those with duplicate references; and those that, after reading the titles and abstracts, did not fit within the limited theme.

A manual search was also conducted, including eligible studies by reading the reference lists of the included articles. After extracting the studies, a flowchart was used (Figure 1), where of the 509 articles derived from the preliminary search strategies, 87 were excluded due to duplicate works, and 395 because of the exclusion criteria established in this study. After reading the abstracts, 11 were excluded, and in the end, 10 studies were selected and included for a full read.

## 3. Results

### 3.1. Nutritional Assessment Methods: Ultrasound

The direct assessment method is considered one of the most accurate for assessing body composition. But because it involves dissecting cadavers, it is not feasible, and there is no data in the literature using this method in children. Among the methods of nutritional assessment in pediatrics, indirect assessment methods are preferable. These include imaging methods: Computed Tomography (CT), Dual Energy X-ray Absorptiometry (DEXA), Magnetic Resonance Imaging (MRI), and ultrasonography. There is a good correlation between the direct method and imaging tests, particularly CT and DEXA, which are considered the most widely used and recommended methods. However, their high cost is an obstacle to their use in clinical practice. Therefore, ultrasonography is an effective and more affordable alternative [13].

Regarding nutritional status diagnosis, the most commonly used methods for assessing body composition in pediatrics are anthropometry, including measurements such as weight, height, circumferences, and skinfold thickness. Body mass index (BMI) can be calculated from weight and height, and is a tool widely used in pediatrics to diagnose nutritional status. Other measurements, such as triceps skinfold thickness, often measured in conjunction with arm circumference to determine fat and muscle reserves, are widely used because they can be performed at the bedside and are inexpensive [13]. Calf circumference is also a low-cost, easy-to-use tool that demonstrates the decrease in lean mass in the first weeks of treatment of critically ill pediatric patients [14]. In addition to these measures, other objective markers can be used in nutritional assessment, such as handgrip strength and laboratory biomarkers, including serum albumin, transferrin, hemoglobin, and lymphocyte count, which provide complementary information on the child’s nutritional and immune status [7].

The use of rectus femoris muscle thickness assessed by ultrasound has been shown to be an excellent predictor of muscle mass loss in pediatric [15], adult, and elderly hospitalized patients. The tool is considered feasible and easy to apply for all healthcare professionals as a method for both assessing and monitoring muscle mass. Several studies have demonstrated its great potential for linear assessment at the bedside of patients with sarcopenia in the ICU [15,16,17]. To measure, the patient must lie on his back with hips extended. A mark is made on the leg to be evaluated, gel is applied, and the transducer must be placed perpendicular to the front of the thigh [12].

The decrease in the thickness of the quadriceps femoris is an intense early phenomenon that occurs in most critically ill children [18]. Monitoring nutritional behavior through anthropometric assessments, with ultrasound being one of the methods, is relevant and helps critically ill pediatric patients [15]. The change in muscle mass can be easily estimated using ultrasound measurement to enable early detection during the stay in the PICU in sedated or cooperative children [18].

Due to the harmful effects of sarcopenia and malnutrition, ultrasound measurements are extremely useful, as they provide ICU professionals with accurate information on measurements, enabling them to make decisions regarding muscle mass loss in critically ill patients and the treatment to be offered [19].

#### 3.1.1. General Characteristics of Ultrasonography and Methods Used for Body Composition Assessment

Ultrasound uses sound waves with frequencies above 20,000 Hz, which are imperceptible to the human ear. The transducer, which is in contact with the skin, converts electrical energy into sound waves. The image is formed by analyzing the difference between the waves emitted and those reflected by the tissues. The higher the frequency, the shorter the wavelength and, consequently, the shallower the penetration depth, but with higher image resolution [20]. When the ultrasound beam contacts the tissue interface, it is partially reflected back to the transducer as an echo. Thus, the transducer has the dual function of transmitting and receiving data. The echoes are transformed into signals so that the transducer can process each wave, with its strength represented by a point and its depth represented by the received echo. In other words, the principle of ultrasound imaging is the reflection of waves that are in the path of the beam [21].

Acoustic impedance is the result of tissue density and acoustic velocity. The acoustic impedances of fat and muscle are similar, and there is a weaker echo for the fat–muscle interface than for the muscle-bone interface. The amplitude of the echoes is represented by reflections, with strong echoes appearing in white and weaker echoes in grayscale, and no echo appearing in black. This results in a two-dimensional grayscale image with white edges for skin–subcutaneous fat and fat–muscle interfaces. Ultrasound transducers vary in mode and frequency. Mode A, or amplitude mode, relies on a beam to scan tissue discontinuity and results in a peak. Mode B, or brightness modulation, uses a linear array and produces a two-dimensional image combined with mode A signals [20,21]. The acoustic impedance of adipose tissue and muscle are similar, presenting a weaker echo between adipose tissue and muscle than between muscle and bone or bone and adipose tissue. That represents a disadvantage during interpretation, thus requiring technical and specific knowledge to interpret the results [22].

A systematic review conducted with the aim of standardizing the use of ultrasound in muscle mass assessment identified a wide variety of equipment configurations used in the studies analyzed. Among the systems employed, the following stand out: SonoSite, GE Healthcare, Siemens, Mindray, Philips, Toshiba, Hitachi Aloka Medical, Esaote, Fukuda Denshi, Hewlett-Packard, and Telemed. All studies used B-mode for image acquisition, except one that used a linear transducer. The frequency of the transducers ranged from 3 to 15 MHz. Although not all studies reported the use of auxiliary software for image analysis, those that cited the following programs: Image J, Matlab (The MathWorks, Inc., Natick, MA, USA), and Photoshop (Adobe, Adobe Systems Incorporated, San Jose, CA, USA) [23].

Regarding the studies included in this review, it was observed that the frequency of the transducers used varied between 4.2 MHz and 13 MHz. About measurement protocols, we also identified discrepancies: some studies performed three measurements (triplicates), adopting the average of these measurements as the representative value [24,25,26,27,28,29]. On the other hand, some authors opted to take four measurements, corresponding to the different muscles that make up the quadriceps femoris, using the average of these values as a reference [12,18,30]. In addition, in two studies, two measurements were taken in the transverse plane and two in the longitudinal plane to calculate the average thickness of the quadriceps femoris [18,28]. The probe was always positioned perpendicular to the skin or thigh, with an abundant amount of transducer gel, and only two studies used maximum compression for the measurement [28,29], while the others used minimum compression. Regarding patient positioning, all studies used the supine position with the hip in extension, neutral knee rotation, and muscles completely extended and relaxed as a protocol.

After reviewing several studies [12,18,24,25,26,27,28,29,30,31] that used ultrasound to assess muscle mass in sarcopenia, we found that, in relation to patient positioning, most studies used the dominant side for assessment. The positioning depended on the targeted muscle, but when in the prone or supine position, the hips and shoulders were always aligned without external or internal rotation and with full extension of the limbs. Regarding the measurement of the quadriceps, which is composed of four muscles: rectus femoris, vastus medialis, vastus intermedius, and vastus lateralis, most were evaluated in the supine position in full extension, but there are also reports of evaluation in the sitting position with hips at 90º and knees at 60º. The most commonly used anatomical landmark was located 2/3 of the distance between the anterior superior iliac spine and the proximal edge of the patella for each of the four quadriceps muscles.

All studies used the femoral quadriceps measurement, but some also used other thickness measurements such as the biceps brachii, triceps brachii, tibialis muscle, gastrocnemius, and diaphragm [12,18,24,25,26,27,28,30]. Regarding the quadriceps femoris, Vallea et al. [18] standardized the measurement at the widest part of the thigh; however, Johnson et al. (2018) [24], Montoro et al. (2023) [26], and Oliveira et al. (2023) [27] considered the measurement to be half the distance from the anterior superior iliac spine to the upper edge of the patella. In the studies by Hoffman et al. (2021) [12], two of five distances from the anterior superior iliac spine to the upper patellar border were considered. And Figueiredo et al. (2021) [28], Gehad et al. (2022) [30], and Mohamed et al. (2025) [29] considered the long axis of the anterior thigh and 2/3 of the anterior superior iliac spine to the upper edge of the patella.

To calculate thickness, the measurement obtained vertically in the image from the outer cortex of the femur to the inner edge of the muscle fascia was considered. Most studies defined muscle atrophy as a 10% decrease in thickness compared with previous assessments. From the initial to the final assessment, muscle loss can reach 13% [12,18,24,25,26,27,28,29,30,31]. It has been shown that increasing age, cumulative energy and protein deficiency, increased C-reactive protein levels, exposure to neuromuscular blockers, and longer stays in the PICU were associated with more severe muscle mass loss [26].

None of the studies analyzed established a specific cut-off point for muscle thickness considering age group or gender. However, Valla et al. (2017) [18], in a study of 35 children aged 0 to 15 years admitted to the PICU, reported an average quadriceps femoris muscle thickness of 2.25 cm at admission. Montoro et al. (2023) [26], in pediatric patients undergoing cardiac surgery with respiratory failure or brain injury, described a median of 1.56 cm, plus a 13% percentage reduction during follow-up.

Oliveira et al. (2023) [27], in a sample of 13 children aged between 1 month and 12 years who remained on mechanical ventilation for at least 24 h, found an average thickness of 1.85 cm. Souza et al. (2018) [31] studied 39 children and teenagers with cystic fibrosis and compared them with a control group of 45 healthy children, with mean ages of 13 and 12.9 years, respectively. The authors observed an average quadriceps femoris thickness of 2.4 cm in the cystic fibrosis group and 2.5 cm in the control group. Consistently, all studies reported a reduction in muscle thickness greater than 10% during hospitalization [12,18,24,25,26,27,28,29,30,31], and Montoro et al. (2023) [26] highlighted an average daily loss rate of 1.57%. Mohamed et al. (2025) [29] found significant muscle atrophy comparing measurements taken on the third and seventh days with those taken upon admission to the PICU.

Muscle echogenicity has been proposed as a parameter indicative of muscle quality, making it a potentially accessible tool for evaluating critically ill children. However, it is not yet possible to accurately determine the degree of muscle changes, since processes such as necrosis may not be well differentiated by variation in echogenicity. Furthermore, there remains a need to elucidate how echographic changes correlate with muscle strength and function [13].

Montoro et al. (2023) concluded that for pediatric patients on mechanical ventilation, ultrasound is an accurate and reliable method capable of identifying skeletal muscle atrophy, which is strongly associated with the use of muscle blockers. Monitoring through this tool allows assessing muscle mass and provides information about nutritional status [26].

Although most available studies focus on the use of ultrasound in pediatric patients admitted to the ICU, its application can also be extended to the wards. The feasibility and safety of the method have already been demonstrated in this population, which reinforces its potential as a tool for monitoring muscle mass in different pediatric clinical settings, even in the absence of standardized protocols.

#### 3.1.2. Challenges of Using Ultrasound in the Assessment of Pediatric Patients

Under an ultrasound examination, children may not be fully cooperative, and for this reason, acquiring images with muscle relaxation can be challenging [28]. That reinforces the fact that movement or changes in position at the time of evaluation can cause changes in muscle thickness [27]. The level of minimum or maximum pressure can significantly alter the reading of the results, as well as the result of muscle mass loss [28].

It is noticeable that the small anatomical dimensions present in children make it difficult to classify ultrasound muscle changes with certainty. That difficulty arises from the broad age range among pediatric patients, which makes it challenging to compare average muscle thicknesses, as these measurements vary significantly among infants, children, and teenagers [28]. Comparing studies with different age groups, significant variations in results are observed. Hence, comparing studies with different age groups shows significant variations in results are observed, infants, children, and teenagers [28]. For example, Mohamed et al. (2025) [29] evaluated children with an average age of 14 months, finding an average muscle thickness of 0.97 ± 0.40 cm (ranging from 0.43 to 2.5 cm). In the study by Valla et al. (2017) [18], the median age was 47 months (ranging from 5 to 126 months) and the median weight was 20 kg (7.8–29.6 kg), with an average femoral quadriceps muscle thickness of 2.25 cm. These data show how age directly influences the interpretation of muscle thickness measurements obtained by ultrasound.

The presence of acute or chronic pathologies is another factor that can interfere with read ultrasound results [27], as they cause changes in muscle echo intensity, making it difficult to visualize the bone echo for the accurate assessment of muscle thickness [25]. The presence of edema is one of the causes that can complicate echo visibility, as increased capillary permeability, typical of severe inflammatory processes, favors the extravasation of fluids from the intravascular compartment to the interstitium, leading to the formation of edema. The reduction in the kidney’s ability to excrete sodium and water, coupled with increased fluid retention, further aggravates swelling. Hydration is also a factor that must be considered since it directly impacts the accuracy of muscle loss assessment [32].

It is known that in critically ill children, fluid overload is a bad sign, as it causes changes in body weight, resulting in a positive balance. Fluid accumulation, even if it does not directly alter muscle mass measurements, compromises the accuracy of the assessment and may lead to underestimation or overestimation of the magnitude of protein depletion, due to the difficulty in visualizing the ultrasound echo [18,28]. It also causes negative clinical outcomes, including longer mechanical ventilation time due to pulmonary edema formation, increased incidence of acute kidney injury, prolonged ICU stay, and elevated morbidity and mortality rates [32].

An additional challenge in ensuring the accuracy of muscle assessment by ultrasound is the level of training of the examiner. Hoffman et al. (2021) [12] demonstrated a significant correlation between sarcopenia, assessed by ultrasound in critically ill patients, and protein-calorie deficiency, highlighting the influence of malnutrition on muscle mass reduction. Furthermore, studies that analyzed intraoperator reproducibility and reliability reported positive results, indicating good consistency of measurements when performed by previously trained evaluators [12,18,27,28].

The reliability of evaluators can be verified by tests, such as the Bland–Altman method, that calculates the relative mean difference between two repeated measurements performed by the same operator [18]. The intraclass correlation coefficient was also used to determine the level of agreement between measurements, classifying them as poor (<0.21), fair (0.21–0.40), moderate (0.41–0.6), good (0.61–0.80), and very good (0.81–1.00). A value below 0.5 indicates low reliability, between 0.5 and 0.75 indicates moderate reliability, between 0.75 and 0.9 indicates good reliability, and above 0.9 indicates excellent reliability [12,25,27].

The studies found sufficient and satisfactory results for repeatability and inter-operator reproducibility, suggesting that with adequate training, the technique has good reproducibility and can be used accurately to assess changes in muscle thickness by different evaluators in the PICU environment [12,18,25,27,28].

Low muscle mass and strength can negatively affect metabolic and bone health before adulthood [15^A^]. That demonstrates the importance of measuring and monitoring muscle mass and function from childhood to adulthood. With constant monitoring, it is possible to identify relevant strategies to improve muscle mass and strength [33]. Muscle thickness measurement accurately assesses muscle atrophy in acquired weakness in the ICU, so evaluating the smallest quantitative changes in muscle mass over the course of ICU stay is essential for a better prognosis [25]. The population studied undergoes significant muscle mass loss during hospitalization, especially in the first few days, which is an intense event in most severely ill children, with an estimated 10% showing muscle atrophy [12,24,26].

However, ultrasonography is a method that has not yet been validated and standardized in children, with no specific cut-off point, requiring a standardized approach for assessing the average thickness of the quadriceps femoris, as it enables the standardization of findings regarding this measurement, allowing routine use in pediatric practice [12,23,28]. It is essential to standardize a rigorous protocol for identifying muscle changes in pediatric patients during hospitalization in the PICU [25].

Among the ten studies (Table 1) included in this review, eight were conducted in PICUs with patients undergoing mechanical ventilation and sedation. Two studies were in patients not on mechanical ventilation, one of them admitted to a pulmonology ward, and compared both with a control group of healthy patients not hospitalized, and the other was on patients with nephrotic syndrome treated at an outpatient clinic, comparing both with a control group of non-hospitalized healthy patients. Regarding the type of study, four were prospective observational, three were prospective cohort, one was cross-sectional observational, and two were cross-sectional controlled [12,18,24,25,26,27,28,29,30,31].

Ultrasound assessment has proven to be an effective tool for identifying this loss, revealing a significant association between skeletal muscle reduction and factors such as prolonged mechanical ventilation and deficits in caloric and protein intake. Specifically, individuals affected by traumatic brain injury (TBI) showed a higher degree of muscle atrophy, accompanied by elevated levels of C-reactive protein (CRP), greater exposure to neuromuscular blockers, and increased length of stay in the PICU—reinforcing the severity of the systemic impact of TBI in these patients [12,18,24,25,26,27,28,29].

Complementing these findings, Ventura et al. (2021) [33,34] highlight other factors that aggravate the deterioration of nutritional status in critically ill children, such as the presence of preexisting chronic diseases, age under two years, fluid overload in the first 72 h of hospitalization, and hypoalbuminemia. These elements not only compromise nutritional status but are also associated with longer hospital stays in the PICU, reinforcing the need for an early and individualized nutritional approach.

This study identified important limitations related to the lack of standardization of measurements, the heterogeneity of the age groups evaluated, and the different forms of analysis of the results. Wide variation in muscle thickness values was observed due to differences in age among participants, which makes direct comparison between studies difficult. In addition, some authors chose to present their findings exclusively in terms of percentage muscle mass loss, limiting the uniformity and comparability of results.

Representation in absolute values, such as measurements in centimeters, enables the future definition of more precise cut-off points according to each age group. On the other hand, the use of relative values in percentages has the potential to highlight progressive muscle reductions, since studies that employed this approach identified atrophy starting at 10% loss [26], in addition to allowing the estimation of the daily rate of reduction. Thus, the integration of both metrics can provide a more comprehensive and accurate assessment of muscle monitoring.

Table 1 summarizes the information regarding the general characteristics of ultrasonography and the parameters used to assess body composition in pediatric patients as described.

## 4. Conclusions

The nutritional assessment of pediatric patients is challenging due to the complexity of care and limitations in methods for assessing body composition and muscle reduction. In this scenario, ultrasound emerges as a promising alternative, as it is a noninvasive method that is easy to apply at the bedside and provides practical information on body composition in this population, establishing itself as a complementary tool for monitoring and assessing nutritional status.

Based on our review, we observed that different models of ultrasound devices are used, which may be portable or not; however, most use mode B. Regarding the frequency used, there is still no standard value, but the range used was up to a maximum of 13 MHz. The main thickness used to assess skeletal muscle loss in pediatric patients was the femoral quadriceps, used in all studies.

Our study concludes that the use of ultrasound to assess skeletal muscle mass reduction in pediatric patients undergoing ICU is accurate and allows for the early detection of atrophy. At present, there is no standard protocol for measurement and analysis of results, lacking reference values, cut-off points, and specific averages for this population. Therefore, further studies are needed to develop a standardized protocol for muscle ultrasound assessment, adopting uniform criteria and well-defined age groups, to standardize the use of ultrasound and establish appropriate reference values and cut-off points for the pediatric population.

One of the challenges encountered in evaluating this type of patient is the wide range of ages, which makes it difficult to compare the average thicknesses found, since the values vary significantly from infants to 18-year-old adolescents. Therefore, further studies using similar criteria, with more defined age ranges, are still needed in order to evaluate an average standard and cut-off point for assessing muscle thickness by ultrasound in the pediatric population.

Currently, there is still no standardized protocol for measuring and analyzing results, including reference values, cut-off points, and specific averages for this population. Based on the studies reviewed, we suggest a minimum operating protocol: patient in supine position with hip in extension; target muscle: quadriceps femoris; probe positioned perpendicular to the skin, with minimal compression; the use of a linear transducer with a frequency between 4.2 MHz and 13 MHz; and measurements taken in triplicate, using the average of the values obtained. Future studies are needed to validate this protocol, establish uniform criteria, and define appropriate reference values for the pediatric population. In addition, future research, preferably multicenter and with more robust samples, is needed to validate this protocol and investigate the correlation between muscle thickness and functional outcomes, such as muscle strength, mobility, and clinical outcomes, in order to evaluate and expand the applicability of the method.

## Figures and Tables

**Figure 1 diagnostics-15-02472-f001:**
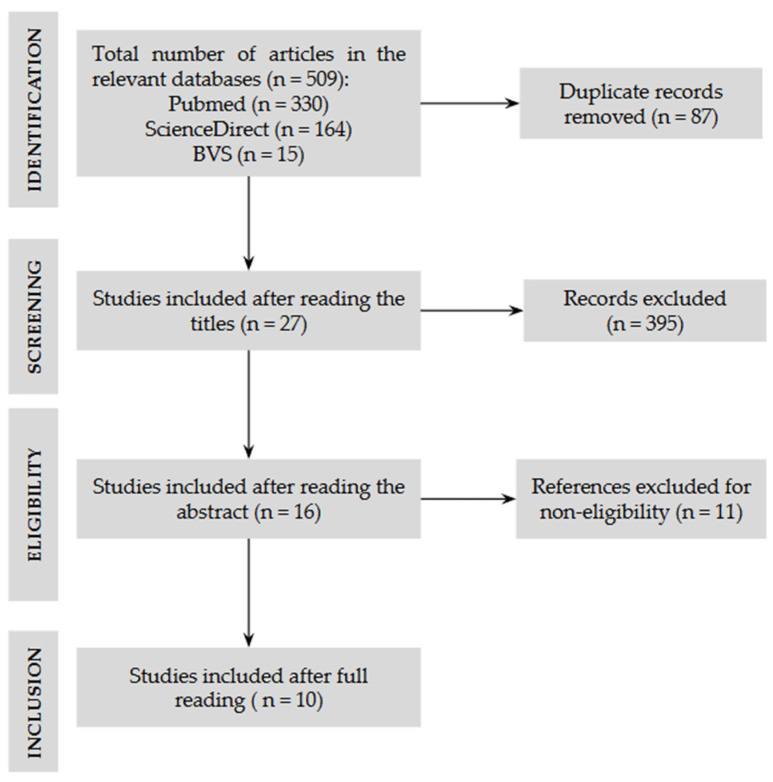
Flow diagram of literature selection process.

**Table 1 diagnostics-15-02472-t001:** Ultrasound parameters and results in the assessment of body composition in pediatric patients.

Author (Year)	USG Model	Parameters Evaluated	Key Findings	Transducer	Location/Measurement Point	Muscle Mass	Thickness
Valla (2017) [18]	Vivid S6/SonoSitand EDGE	Age, sex, weight, BMI, energy and protein intake, duration of MV, cachexia	QF thickness reduction: 9.8% (day 5) and 13.3% (final); good inter-rater reliability	Linear, 9–13 MHz	Thigh: widest portion measured from this point to the upper tip of the patella, perpendicular probe, minimal compression, supine position, 4 measurements (average), unilateral	QF thickness: femoral cortex to fascia	Median: 2.25 cm
Hoffman (2021) [12]	GE Logiq S8/Nextgen	Water balance, weight, nutritional intake	Reduction in QF thickness associated with caloric and protein deficiency; good reliability	Linear, 10 MHz or 4.2–13 MHz	Thigh: 2/3 anterior superior iliac spine of the patella, perpendicular probe, minimal compression, supine position, 4 measurements (average), unilateral	QF thickness: femoral cortex to fascia	2.4 ± 0.56 cm
Johnson (2018) [24]	SonoSitand Edge II	Electrical impedance myography, clinical and demographic variables	Reduction: diaphragm 11.1% and QF 8.62%; 83% muscle atrophy; age and TBI → greater loss	Linear, 6–13 MHz	Diaphragm, biceps, QF, tibial: defined anatomical locations; minimal compression; supine position; 3 measurements (average), unilateral	Atrophy defined as ≥10% reduction; increased fat and reduced muscle quality	QF: ↓8.62% or 1.5%/day
Ng (2019) [25]	SonoSitand Edge II	Clinical and demographic variables	Correlation between muscle thickness and age and weight; atrophy: >13% limbs and >38% diaphragm	Linear, 6–13 MHz	Biceps, QF, tibial: maximum diameter; diaphragm: intercostal window; minimum compression; supine; 3 measurements (average), unilateral	Thickness: perpendicular to fascia/bones; diaphragm: between pleural layers	QFMT: 2.19 ± 0.89 cm
Montoro (2023) [26]	SonoSitand M-Turbo	Age, sex, BMI, energy and protein intake, duration of MV, neuromuscular blockers	QF thickness reduction: average 13% (1.57%/day); predictors: age, blockers, greater inflammation	Linear, 12 MHz	Thigh: midpoint of the anterior superior iliac spine of the patella; perpendicular probe; minimal compression; supine position; 3 measurements (average), unilateral	Thickness: femoral cortex to fascia	QFMT: 1.56 cm (median)
Oliveira (2023) [27]	InnoSight	Anthropometric and nutritional measurements	Reliable USG for muscle changes in UTIP	Linear, 4–12 MHz	Biceps, QF: anatomical locations; minimal compression; supine; 3 measurements (average)	Thickness defined by bone landmarks; assessment of muscle mass	Bíceps: 1.14 cm; QF: 1.85 cm
Souza (2018) [31]	DP 6600 Laptop	Anthropometric measurements, skinfold thickness, nutritional changes	USG correlated with skinfold thickness in the assessment of body fat	Linear, 7.5 MHz	Triceps, QF, gastrocnemius: anatomical landmarks; minimal compression; supine; unilateral	Thickness: distance between bone and muscle boundaries; subcutaneous fat	Triceps: 1.4 cm; QF: 2.4 cm; Gastrocnemius: 1.7 cm
Figueiredo (2021) [28]	Vivid Q	Anthropometry and nutritional supply	Significant reduction in QFMT in both subgroups; correlation with protein deficiency	Linear, 5–13 MHz	Thigh: 2/3 anterior superior iliac spine of the patella; maximum compression; supine; 3 measurements (average), unilateral	Thickness: from the upper margin of the femur to the deep fascia	Subgroup 1: 0.65 ± 0.23 cm; Subgroup 2: 0.63 ± 0.22 cm
Gehad (2022) [30]	Philips HD7xe	Age, sex, BMI, energy and protein intake, renal function	Smaller QRF and QVI muscles in the NS group; greater subcutaneous fat.	Linear, 7.5 MHz	Thigh: Measurement at the midpoint and at the border between the upper 2/3 and lower one-third of the line between the upper pole of the patella and the anterior superior iliac spine.	Maximum thickness measured from the inner edge of the muscle to the femur (QVI) and from the subcutaneous layer to the inner edge (QRF).	QRFT): NS group: 1.088 cm Control group: 1.27 cm (QVIT): NS group: 0.999 cm Control group: 1.138 cm.
Mohamed (2025) [29]	GE Logiq S8	Age, gender, BMI, medical history, diet, use of supplements, anthropometry, duration of VM, tests.	Significant reduction in QFMT on days 3 and 7 after admission, correlation between nutritional intake and QFMT thickness, muscle atrophy associated with longer hospital stays and unfavorable outcomes.	Linear 11 MHz	Thigh: 2/3 anterior superior iliac spine of the patella; perpendicular probe, maximum compression, supine position, 3 measurements (average), unilateral	Thickness: distance between the upper edge of the muscle and the cortex of the femur.	QFMT: 0.97 ± 0.40 cm (0.43–2.5 cm)

USG: Ultrasound, BMI: Body Mass Index, QF: Quadriceps Femoris, TBI: Traumatic Brain Injury, QFMT: Quadriceps Femoris Muscle Thickness, MV: Mechanical Ventilation, QRF: Quadriceps Rectus Femoris, QVI: Vastus Intermedius, NS: Nephrotic syndrome, QRFT: Quadriceps Rectus Femoris Thickness, QVIT: Quadriceps Vastus Intermedius Thickness.

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
