# Peer review of "Use of Ultrasound for Body Composition in Assessment in Pediatric Patients: Are There Still Challenges?"

_diagnostics, 2025, doi:10.3390/diagnostics15192472_

Round 1
Reviewer 1 Report
Comments and Suggestions for Authors
Introduction:
Comment 1.
Please rephrase or explain “This literature review is timely because muscle mass loss during hospitalization is still present.” What do you mean by “still”
Comment 2.
Line 96 “less accurate methods”. Please analyze
Comment 3
Line 96 “It is a measurement that is not affected by water balance”, however you comment that fluid overload and edema do complicate assessments i.e. “Hydration is also a factor that must be considered since it directly impacts the accuracy of muscle loss assessment” and “Fluid accumulation, although it may not directly impact muscle mass measurements, compromises the accuracy of the assessment and may lead to underestimation or overestimation of the magnitude of protein depletion”. It remains unclear to me if muscle mass measures are affected by water imbalance.
Results
Comment 3
Please rephrase “Anatomical landmarks from the greater trochanter to the proximal edge of the patella for the four respective quadriceps muscles.”
Comment 4
Please rephrase and clarify “Regarding the quadriceps femoris, standardized the measurement at the widest part of the thigh”. Who?
Comment 5
Line 330-332. How is this relevant to examiners training?
Author Response
"Please see the attachment."

Reviewer 2 Report
Comments and Suggestions for Authors
Dear Authors,
The manuscript (diagnostics-3823853) is well-written, clear, and relevant. However, it remains closer to a narrative review than a systematic one. It is valuable in drawing attention to the importance of ultrasound in pediatric critical care, but its impact could be strengthened with:
- greater methodological rigor (PRISMA compliance, study quality analysis),
- deeper quantitative insights,
- and the proposal of a practical clinical protocol.
Therefore, the reviewer suggests the major revision for this manuscript.
Strengths
- Relevance of the topic
- The use of ultrasound to assess body composition and muscle loss in pediatrics is timely and not yet standardized, so this review addresses an important gap.
- Focusing on critically ill pediatric populations (e.g., ICU patients) is original and clinically relevant.
- Bibliographic methodology
- The review specifies the time frame (2015–2025), databases used (PubMed, VHL, Science Direct), and DeCS/MESH descriptors.
- Inclusion of both prospective and retrospective studies, as well as reviews and meta-analyses, shows a comprehensive approach.
- Clear results
- Provides a summary of ultrasound measurement techniques (rectus femoris, quadriceps).
- Detailed description of technical variables (frequency, patient positioning, number of measurements).
- Highlights clinical correlations with outcomes such as prolonged mechanical ventilation, nutritional deficits, inflammation, and exposure to muscle relaxants.
- Critical discussion
- Accurately identifies current limitations: lack of standardization, high variability in protocols, absence of pediatric cut-offs, challenges related to age and body size.
- Notes practical issues (child cooperation, edema, operator experience).
Weaknesses
- Limited methodological depth
- The quality of the included studies is not assessed (e.g., no bias assessment tools such as PRISMA or ROBIS).
- Despite inclusion/exclusion criteria, the review feels more narrative than systematic.
- Heterogeneity of studies
- The synthesis is descriptive but lacks quantitative comparison (no meta-analysis or numerical contrasts of muscle thickness reductions).
- Restricted generalizability
- The focus spans very different conditions (cystic fibrosis, nephropathies, TBI, mechanical ventilation), making it difficult to derive general conclusions.
- No concrete protocol proposal
- The article concludes with the need for standardization but does not propose operational guidelines or even a minimal methodological scheme.
Suggestions for Improvement
- Improve methodological rigor
- Clarify whether the review follows PRISMA guidelines (checklist and flow diagram).
- Include a critical appraisal of the methodological quality of included studies.
- Enhance quantitative analysis
- Provide a comparative table of average muscle thickness values across age groups.
- Use graphical representation of reductions (e.g., % muscle loss per day of ICU stay).
- Expand practical clinical implications
- Discuss how ultrasound can currently be used in pediatric wards despite the lack of standardized protocols.
- Highlight the potential utility of relative values (e.g., % change from baseline) rather than absolute cut-offs.
- Propose a standardization framework
- Suggest a minimal operative protocol (supine position, target muscle = quadriceps femoris, number of measurements, probe compression standard).
- Identify priority research areas (e.g., multicenter validation, correlation with functional outcomes).
Author Response
"Please see the attachment."
